# Wnt Signaling Pathway in Tumor Biology

**DOI:** 10.3390/genes15121597

**Published:** 2024-12-13

**Authors:** Sabina Iluta, Madalina Nistor, Sanda Buruiana, Delia Dima

**Affiliations:** 1Department of Hematology, Iuliu Hatieganu University of Medicine and Pharmacy, 400347 Cluj Napoca, Romania; iluta.sabina@yahoo.com; 2Medfuture Research Center for Advanced Medicine, Iuliu Hatieganu University of Medicine and Pharmacy, 400347 Cluj Napoca, Romania; 3Department of Hematology, Nicolae Testemitanu University of Medicine and Pharmacy, 2004 Chisinau, Moldova; sanda.buruiana@usmf.md; 4Department of Hematology, Ion Chiricuta Oncology Institute, 400015 Cluj Napoca, Romania

**Keywords:** Wnt pathway, cancer stem cells, tumor biology

## Abstract

Relapse and metastasis are the major challenges that stand in the way of cancer healing and survival, mainly attributed to cancer stem cells (CSCs). Their capabilities of self-renewal and tumorigenic potential leads to treatment resistance development. CSCs function through signaling pathways such as the Wnt/β-catenin cascade. While commonly involved in embryogenesis and adult tissues homeostasis, the dysregulation of the Wnt pathway has direct correlations with tumorigenesis, metastasis, and drug resistance. The development of therapies that target CSCs and bulk tumors is both crucial and urgent. However, the extensive crosstalk present between Wnt and other signaling networks (Hedgehog and Notch) complicates the development of efficient long-term therapies with minimal side-effects on normal tissues. Despite the obstacles, the emergence of Wnt inhibitors and subsequent modulation of the signaling pathways would provide dynamic therapeutic approaches to impairing CSCs and reversing resistance mechanisms.

## 1. Introduction

Wnt, Notch, and Hedgehog (Hh) are developmental signaling pathway with vital importance in regulating cell fate decisions like tissue growth and homeostasis or stem cell renewal, migration, or differentiation [1]. These processes are indispensable for tissues like blood, brain, skin, lungs, and intestine. However, with such fundamental implications in basic cellular processes, these signaling networks are often prone to malfunctioning into human malignancies. The ongoing study of cancer has brought important observations such as the constant presence in tumors of a subpopulation of cells with stem-like properties, notoriously known as cancer stem cells (CSCs) [2].

Various key signaling networks, mainly Hh, Notch, and Wnt, appear to influence the fate and behavior of CSCs. This, indeed, presents both challenges and opportunities, as targeted antagonization of such pathways may give rise to promising anti-cancer treatments. In the last decade, CSCs were recognized in various cancers. The presence of tumor heterogeneity and small populations of cells with stem-like properties have been established in the majority of malignancies, although debates about their origins and plasticity continue [3].

These CSCs share a number of similarities with embryonic or tissue stem cells and are often characterized by the sustained activation of well-conserved signaling pathways such as Notch, Hh, and Wnt. In addition, their slow cell cycle due to stemness properties and intrinsic resistance to chemotherapy and radiotherapy pose difficulties in targeting them [4,5]. Such factors give rise to innovative treatments that concentrate on these pathways to regulate the proliferation, survival, and differentiation of CSC. Lately, developing novel strategies to target these tumor compartmentalized cells represents a heterogeneous blend of therapeutic agents to minimize compensatory mechanisms by which tumors escape. Future plans aim to focus on dealing with resistant mechanisms pertinent to CSCs and further drive clinical development and optimize clinical outcomes in cancer patients [6]. The present work aims to bring clarity to the cellular mechanisms controlled by the above-mentioned signaling pathways and discuss the potential success in targeting them as antitumoral therapies.

## 2. Wnt/β-Catenin Signaling, Embryogenesis, Cell Cycle Regulation, and Cancer Development

Wnt/β-catenin is a major signaling pathway that plays fundamental roles in understanding the complex mechanisms of cancer development, regulation, and progression. It has crucial implications in processes like cell development, growth, and differentiation, but has also been shown to influence cancer pathogenesis and cancer stem cells (CSCs) [6].

The Wnt pathway was first discovered in 1982 by Nusse et al., as an attempt to investigate the genetic modifications involved in the development of breast cancer in mice [7]. The infection of C3H murine models with the mouse mammary tumor virus (MMTV) led to the observation of a proviral insertion mutation in chromosome 15, at a specific locus, which led to the first description of the int-1 (integration 1) gene, an oncogene activated by the retroviral infection. At the same time, independent studies in *Drosophila melanogaster* elucidated the wingless (wg) gene as the core component of embryonic segment polarity [8,9]. Further investigations linked wg with int-1 as homologous genes and came up with the collective term Wnt-“wingless”, undoing the contrived criteria that initially coined “in1” into int for naming [10,11].

Then, it became active in embryogenesis when its role was played in regulating body anterior–posterior axis formation, enabling the epithelial to mesenchymal transition of cells during gastrulation, organogenesis, cell fate specification, and tissue patterning in development. For example, in lung physiological development, the inhibition of the Wnt/β-catenin pathway impacts the differentiation of proximal stem cells into airway epithelial cells, while its activation results in distal epithelial cell specialization [12]. In the skeletal system, Wnt signaling affects the proliferation and differentiation of mesenchymal stem cells and osteoclasts, contributing to bone formation and maintenance of mass and strength [13]. The Wnt pathway also influences the development of the neural tube, heart, and kidney through mechanisms that sustain and enable the proper population of stem cells to differentiate into the mature and functional cells and tissues.

With the later discoveries that underpinned Wnt, a stimulating link with the cell cycle had been established because, through β-catenin stabilization, it influences the transcription and expression of key regulatory genes like cyclin D1 and c-myc, essential for the G1/S transition in the cell cycle [14]. In the case of pluripotent stem cells, Wnt signaling os involved in different steps of the cell cycle. First, just like in the case of somatic cells, it promotes cell proliferation by facilitating the G1/S transition, due to cyclin D1 transcription and inhibition of p21 and p27. Proliferation is also stimulated by activation of the highly conserved cyclin Y, with indispensable role in mitosis regulation (G2/M transition) [15].

Structurally, the Wnt signaling axis has been described to consist of several pathways with particular functions, especially in body morphogenesis. The “canonical” pathway [16,17,18,19], so-called due to its reliance on the key protein β-catenin, is well studied. The “non-canonical” Wnt signaling, on the other hand, functions independently of β-catenin and involves pathways such as planar cell polarity, Wnt/Ca^2+^, protein kinase A, the c-Jun N-terminal cascade, and small GTPase signaling (RhoA, Rac1, and Cdc42) [20,21].

Wnt proteins are synthesized in the endoplasmic reticulum of the secreting cell and function as paracrine or autocrine glycoproteins promoting cell growth. After their secretion, these proteins act as ligands for a receptor complex involving a Frizzled family member and a member of the low-density lipoprotein receptor-related protein (LRP), either LRP5 or LRP6. The Wnt/Frizzled/LRP5/6 complex activates kinases that phosphorylate the cytoplasmic tail of LRP, leading to the recruitment of axin, a scaffolding protein, to the complex [22]. Axin is part of a multi-protein complex that includes the tumor suppressor, adenomatous polyposis coli (APC), glycogen synthase kinase 3 (GSK3), and casein kinase 1 (CK1).

When Wnt signaling is non-functional, the complex degrades β-catenin by allowing GSK3β to phosphorylate and mark it for proteolysis. Next, the protein is polyubiquitinated by the ubiquitin ligase complex and subsequently degraded by the 26S proteasome [23]. This process prevents β-catenin from moving to the nucleus and activating target gene expression [24].

When Wnt is activated, the ternary Wnt/Frizzled/LRP complex gathers the Dishevelled (Dvl/Dsh) phosphoprotein to the plasma membrane and blocks the axin-bound GSK3β complex. GSK3β becomes inactivated, stopping cytosolic β-catenin from degradation. β-catenin, once stabilized, accumulates in the nucleus and interacts with transcription factors from T-cell factor lymphocyte enhancer factor (TCF/LEF), promoting the activation of Wnt-targeted genes [25,26], including cyclin D1, vascular endothelial growth factor (VEGF), CD44, c-Myc, matrix metalloproteinase 7 (MMP7), and Runt-related transcription factor 2 (RUNX2) [27]. These genes are linked to cell proliferation, cell cycle progression, growth factor expression, apoptosis, angiogenesis, and the expansion of stem cell states, all of which are crucial traits in some cancer cells. However, the implication of non-canonical Wnt signaling in cancer initiation and progression is yet to be elucidated. This complexity arises because multiple non-canonical pathways can be activated by a ligand or receptor, with either inhibitory or stimulatory effects. Additionally, the lack of specific markers for non-canonical pathways obstructs the precise understanding of their role in embryonic and cancer cells. The Wnt/Ca^2+^ pathway is sometimes seen as an inhibitor of the Wnt/β-catenin pathway, functioning as a tumor suppressor [28]. Depending on the receptor type, the Wnt-5a ligand, a key player in non-canonical signaling, can act as either a proto-oncogene or a tumor suppressor [29]. For instance, Wnt-5a, commonly involved in neural crest cell migration, decreases its expression once cells reach their destination tissue [30]. In melanoma, Wnt-5a was shown to increase protein kinase C (PKC) activity [31] and MMP secretion [32] and, thus, enhance invasiveness in an independent manner from β-catenin-mediated-transcription. The planar cell polarity (PCP) pathway also appears to contribute to colorectal cancer evolution, with components such as VANGL1 promoting increased cell adhesion and invasion. Suppression of VANGL1 counteracts these effects [33], potentially due to its interaction with a PKC inhibitor [34]. In colorectal cancer, histone deacetylase inhibitors have been shown to suppress the clonogenicity and proliferation of cancer cells, both stem and non-stem, potentially through the activation of non-canonical Wnt pathways [35].

Wnt signaling is regulated by natural inhibitors that keep β-catenin levels low within cells. These inhibitors either bind Wnt ligands or Wnt receptors. “Decoy” proteins, such as glycoproteins of the secreted Frizzled-related protein family (SFRP), Wnt inhibitory factor-1, and Cerberus, resemble Wnt receptors and block the signaling pathway. Meanwhile, Dickkopf (DKK) proteins bind to LRP5/6, preventing the formation of the Wnt/Frizzled/LRP5/6 complex and halting Wnt signaling [36,37,38,39].

Increased Wnt signaling often results from deregulations in key pathway components, which vary across cancer types. For example, most colorectal cancers are caused by mutations in the Adenomatous polyposis coli (APC) gene, which usually acts as a tumor suppressor through β-catenin degradation. However, the loss of APC function causes a constant activation of β-catenin, contributing to both the initiation and progression of colorectal cancer [40,41,42]. Mutations in the APC gene appear in nearly 85% of sporadic intestinal cancer cases [43]. Furthermore, APC turn-off has been linked to elevated expression of stem cells markers like CD44, Lgr5, Bmi1, and Musashi1, which support the role of Wnt signaling in maintaining CSC characteristics [44,45]. Around 15% of sporadic colon cancers have showed mutations in the β-catenin gene that make it resistant to degradation, leading to constitutive pathway activation. Such mutations also occur in other cancers, including hepatoblastomas and hepatocellular carcinomas (HCC) [46], and APC and β-catenin mutations have been found together in the same tumor [47].

Wnt signaling is also involved in the epithelial-to-mesenchymal transition (EMT), a key step in metastasis where epithelial cells acquire a mesenchymal phenotype [48]. EMT leads to the loss of E-cadherin, an important part of the of E-cadherin/β-catenin adhesion complex, a form of adherens cell–cell junction. This loss frees β-catenin, increasing its cytoplasmatic levels, promoting the transcription of growth factors in damaged tissues. Alterations in E-cadherin are linked to more aggressive cancer types, seen in malignancies such as esophageal, colon, breast, and prostate cancers [49]. Contrary, Wnt signaling disruption can trigger the reverse process, mesenchymal-to-epithelial transition (MET), highlighting Wnt’s role in maintaining the mesenchymal state [48].

Upregulated Wnt signaling enhances the stemness of both normal and cancer cells. In the intestinal crypt, high levels of nuclear β-catenin are present in stem cells [50], and colorectal cancer cells exhibit a similar genetic profile as the normal crypt progenitors [51]. CSCs are often found near myofibroblasts, which are thought to secrete factors that promote β-catenin-mediated transcription, suggesting the tumor microenvironment (TME) is involved in CSCs clonogenicity [52]. The interactions between Wnt and other mitotic pathways may be essential to understanding CSCs biology, as discussed in the following section.

### 2.1. Crosstalk with Other Signaling Pathways

As Wnt signaling was described to have a pivotal role in tumorigenesis, it has become an attractive therapeutic strategy. However, there are numerous reports that the Wnt/β-catenin cascade is often associated with other two signaling pathways: Hh and Notch [53]. Crosstalk points between these three pathways involve mutual regulation whereby they can either activate or repress each other’s activity in cancer.

The Notch specific ligand, Jagged1, was shown to downregulate Gli2, an Hh effector, and, thus, promotes apoptosis and increases sensibility to drugs, as observed in a study conducted on in vitro and in vivo ovarian cancer models [54]. A separate study showed that Gli1, another Hh effector, was silenced by Hes1, a Notch-specific protein that encodes a basic helix–loop–helix (bHLH) transcriptional repressor, in glioblastoma neurosphere lines and primary human glioblastoma lines [55].

Wnt interacts with both Notch and Hh, to stabilize Gli or repress β-catenin through Gli interactions. Acar et al., 2021, have brought strong evidence that Notch has a repressive effect on Wnt by forming a complex with β-catenin through two proteins, ΔEGF_N1 and NICD (Notch intracellular domain), thus, blocking the transcription and expression of β-catenin [56]. This complements a previous study from 2014, where Collu et al. correlated GSK3β, a member protein of the β-catenin destruction complex, that additionally phosphorylates NICD and enables its translocation in the nucleus for transcription activation [57].

Similarly, Wnt and Hh also work in an alternative manner. Hh enhances β-catenin stability, an activator of Wnt ligand expression, while Wnt inactivates Hh through β-catenin degradation of gli proteins [58]. Collectively, these pathways promote processes like EMT, CSCs maintenance, and therapy resistance, illustrating a cooperative contribution to tumor progression and treatment evasion [6].

As previously noted, the components of the Wnt/β-catenin pathway are often overexpressed in CSCs, contributing to the preservation of their stem-like properties and fostering aggressive cancer behavior by activating anti-apoptotic mechanisms and promoting resistance to therapy. However, these characteristics are not exclusive to the Wnt/β-catenin pathway; other signaling cascades, such as PI3K/AKT and RAS/RAF/ERK, exhibit similar effects. This raises an important question of whether these pathways operate independently or they interact, thus, obstructing attempts to target Wnt signaling.

Both cancerous and non-cancerous models reveal that Wnt/β-catenin, PI3K/Akt, and RAS/RAF/ERK pathways connect at a shared point—GSK3β. In addition to its central role in Wnt signaling, GSK3β acts downstream of tyrosine kinase receptors and regulates its activity. For instance, in cells with elevated PI3K/AKT activity, the inhibition of GSK3β occurs. The phosphorylation of GSK3β by PI3K/AKT, which leads to β-catenin accumulation, triggers the dedifferentiation of non-CSCs into CSCs, enhancing their therapy-resistant and tumorigenic phenotypes. By contrast, AKT inhibition reduces GSK3β activation through decreased phosphorylation, blocking this dedifferentiation process [59]. Similarly, boosting β-catenin-CBP pharmacologically or genetically diminishes stem-like features and inhibits non-stem to stem cell conversion [59]. Notably, this interaction between Wnt and PI3K/AKT signaling is not limited to cancer cells. In human embryonic and intestinal stem cells, PI3K/AKT controls differentiation by activating Wnt/β-catenin signaling via GSK3β [60,61]. In addition to PI3K/AKT, MAPK and PK A, B, and C can also phosphorylate GSK3β, inhibiting β-catenin degradation and promoting cell proliferation, survival, and aggressive behavior [62]. Furthermore, nuclear β-catenin accumulation enhances cyclin D1 mRNA production, while GSK3β inactivation furtherly boosts cyclin D1 levels, which may partly contribute to its protein degradation [63].

When Wnt signaling is blocked, GSK3β phosphorylates other key growth-regulating proteins besides β-catenin, such as TSC2, a regulator of the mammalian target of rapamycin (mTOR). Phosphorylated TSC2 prevents Rheb activation, thereby inhibiting mTOR signaling. On the other hand, Wnt activation inhibits GSK3β by binding to the Axin-bound-GSK3β complex, promoting mTOR signaling and cancer cell proliferation [64]. Targeting this pathway could be relevant in cancer treatment, as studies in murine models have shown that the mTOR inhibitor everolimus reduced cancer mortality and polyp formation. This effect was linked to β-catenin, as β-catenin knockdown decreases mTOR activity in cancer cells [65].

### 2.2. Targeting Wnt Signaling in Cancer

Despite the pivotal role of β-catenin in cancer development and progression, the blocking of this intracellular protein has proved to be challenging due to its lack of enzymatic activity, which renders it as “undruggable” [66]. Nevertheless, the Wnt signaling pathway offers multiple potential alternative targets for anti-cancer therapies, aimed to stop aberrant Wnt-driven transcription (Table 1). Given that Wnt is a key regulator of EMT and VEGF is a target of the β-catenin/TCF complex [67], it is reasonable to propose that Wnt signaling is critical in promoting cancer invasion and spreading. Therefore, anti-Wnt therapies could be valuable in targeting metastasis or stopping cancer cell migration.

However, anti-Wnt therapies need to be approached with caution, as repressing Wnt signaling could adversely affect the maintenance and proliferation of normal stem cells, including hematopoietic and gastrointestinal cells and consequently impair osteoblast differentiation, with negative impact on bone formation [68,69]. Desirably, CSC-specific inhibitors should be developed to minimize the side effects on normal tissues. Recently, novel approached have been proposed to achieve this goal. The development of multi-task nanoparticles was proposed, for a more specific target with improved drug dosing and also the possibility of using drug combination cargo for a more versatile application that would focus on at least one mechanism involved in cancers, unlike free drugs currently used in common therapies. Multiple models are under development with promising results, but the long-term toxicity and off-target events are still unclear [70].

While targeting Wnt in cancer is promising, several challenges hinder its success. The high diversity of Wnt ligands and Frizzled receptors complicates precise targeting. A total of 19 Wnt ligands and 10 Frizzled receptors have been described so far, and the interactions between them are not uniform. For example, Wnt-5a and Wnt-7a can suppress oncogenic Wnt signaling by triggering non-canonical Wnt pathways [18]. Additionally, the role of Wnt in maintaining various cell populations in the body raises toxicity concerns, so that the safety and efficacy of Wnt-targeting strategies are still in the early stages. Even though some Wnt ligands like Wnt-5 and Wnt-7 inhibit Wnt signaling in cancer cells, there are concerns regarding the involvement of Wnt activators in the promotion of tumor progression through other pathways, which are not fully investigated. For instance, while Wnt-5a generally inhibits Wnt signaling, it also enhances invasiveness in melanoma cells [31].

Several Wnt signaling inhibitors have been identified for different types of cancer, including small molecules and natural inhibitors. The Wnt pathway also displays resistance to radiation and cytotoxic drugs in CSCs by increasing DNA damage tolerance [71,72]. In pancreatic cancer, Wnt-mediated chemoresistance is linked to the upregulation of survivin, an apoptosis inhibitor [73], and livin, a caspase inhibitor [74]. Additional mechanisms include increased expression of DNA repair proteins like Mre11 [75] and MDR-1, which encode P-glycoprotein, a membrane transport protein that block chemotherapeutic agents [76]. In osteosarcoma, inhibiting the Wnt/β-catenin pathway sensitized cancer cells to methotrexate, with even greater results after adding a Notch inhibitor [77]. In HCC, sorafenib reduced Wnt signaling, decreasing resistance to cisplatin therapy [78]. Wnt-mediated resistance to cisplatin has also been observed in non-small cell lung cancer (NSCLC).

While the Wnt pathway is well-studied, its complexity presents challenges for direct targeting, prompting recent analyses of inhibitors acting at various stages of the signaling axis for potential selective anti-cancer effects. The main challenges in achieving effective Wnt inhibition arise from a lack of pathway-specific targets and the increasing redundancy of the signaling pathways.

Wnt ligands undergo structural changes prior to their release into the extracellular environment, like their acylation with a palmitoyl group in the endoplasmic reticulum by the acetyltransferase Porcupine. This process is crucial for the secretion and signaling activity of Wnt ligands. The IWP (“Inhibitor of Wnt production”) molecule has been shown to disrupt this process, demonstrating significant effects in gastric cancer [70,79]. A similar mechanism of action appears with other inhibitors as well when observed in in vivo models of breast cancer and head and neck squamous cell carcinoma [80].

As mentioned earlier, due to the intricate interactions between Wnt ligands and their receptors, using Wnt ligands with known inhibitory effects has been met with hesitation. However, natural inhibitors of Wnt signaling may behave differently in cancer cells. For example, overexpression of SFRPs, which serve as decoy receptors and are often silenced in cancer, can inhibit Wnt signaling even when downstream proteins have activating mutations [81]. Similarly, antibodies targeting the Wnt1 ligand have shown inhibitory effects in various cancer cell lines [82,83]. For example, intraperitoneal administration of anti-Wnt-3a antibodies has demonstrated a therapeutic impact in prostate cancer development [84]. Segments of Frizzled (Fzd) receptors have also been shown to enhance Wnt inhibition, as Fzd7 fractions were found to reduce β-catenin-dependent gene transcription in HCC [85]. Recombinant proteins, such as the soluble Wnt receptor composed of the extracellular cysteine-rich domain of Fzd-8 and the human Fc portion, have also shown anti-cancer effects in teratocarcinomas [86].

Blocking the extracellular phase of Wnt signaling may be especially useful in cancers like triple-negative breast cancer or NSCLC, which are thought to harbor autocrine-like Wnt signaling [87,88]. Additionally, natural Wnt inhibitors from the Dkk family, which target the LRP co-receptor, have been shown to induce apoptosis in various cancers [89,90], although the potential invasion-promoting effects of certain Dkk variants in some cancer types are not to be neglected [91,92].

Fzd receptors have also become the focus of new anti-cancer agents. More exactly, antibodies that target Fzd-10 led to the inhibition of cancer growth in a synovial sarcoma cell line [93]. These results promoted a new therapeutic approach, the administration of adjunct radiotherapy, by attaching Yttrium-90 to the anti-Fzd-10 antibodies [94]. Targeting Fzd-7 with other types of antibodies has also shown therapeutic relevance against colon cancer cells, with no effect on normal tissue [95]. The focus on anti Fzd-7 treatment for other types of cancer has also been explored [96]. Interestingly, peptides derived from the extracellular portion of the Fzd-7 receptor have been found to reduce β-catenin-dependent gene expression in HCC [85].

While targeting Dvl may initially seem counterproductive, given its common PDZ motif used to bind Fzd receptors, recent findings suggest that the PDZ-binding domains of Fzd receptors possess unique characteristics that make Fzd–Dvl binding targetable [97]. Various molecular inhibitors of the Dvl PDZ domain have been described, with some showing reductions in cell proliferation in prostate cancer cells [98] and induction of apoptosis in non-small cell lung cancer and melanoma cells [99]. Peptides that target the Dvl-binding domain of Fzd-7 have also induced apoptosis selectively and promoted β-catenin degradation in HCC [100].

Overexpressed β-catenin is phosphorylated and marked with ubiquitin for proteasomal digestion by a destruction complex. Among all the proteins included in this degradation assembly, namely APC, Axin, GSK-3β, and CK1 have a pivotal role in blocking excessive Wnt signaling by facilitating the binding between β-catenin to the rest of the proteins in the complex [43]. The high frequency of APC mutations in colorectal cancer raises the question of whether restoring normal APC levels is possible. Efforts to increase APC concentration through APC-expressing liposomes have shown success in reducing colorectal cancer occurrence in APC-min mice [101]. On the other hand, Axin, another protein in this complex, is regularly degraded by tankyrase, a poly-ADP-ribose polymerase that attaches ADP-ribose polymers to Axin, targeting it for ubiquitination and degradation. Tankyrase inhibitors like XAV939 and IWR-1 have been shown to suppress Wnt signaling in APC-min cells [102], although their toxicity to normal cells may limit their therapeutic use [70]. Other approaches to promoting β-catenin degradation include pyrvinium, which activates CK1 to enhance β-catenin phosphorylation and degradation [103]. In cases where mutations in Axin or APC render inhibition of Wnt ligands or receptors ineffective, targeting the binding between β-catenin and TCF/LEF transcription factors has shown some more success [104,105].

MicroRNAs, a new class of potential antineoplastic agents, may also influence Wnt signaling. For instance, miR-30 variants can suppress tumors in multiple myeloma cases, where their downregulation leads to the upregulation of the β-catenin co-activator BCL-9, thereby enhancing Wnt-related transcription in cancer cells. Restoration of miR-30 activity in stromal cells has been shown to increase drug sensitivity in chemoresistant myelomas [106]. In prostate cancer, miR-320 suppresses Wnt signaling by directly targeting β-catenin mRNA, and its downregulation has been linked to increased stem-like characteristics in cancer cells. Restoring miR-320 expression not only inhibits stem-like features in CD44-overexpressing cells but also reverses EMT-like processes in prostate cancer cells [107].

Wnt signaling is also receptive to several other small molecule inhibitors that have already been approved for different pharmacological effects. For example, non-steroidal anti-inflammatory drugs (NSAIDs) and selective COX-2 inhibitors like celecoxib have been shown to reduce Wnt-dependent transcription and the incidence of FAP-related colon polyps in both mouse models and human patients [108,109,110]. These molecules may also affect Wnt-dependent CSCs, which are thought to contribute to the overall chemoresistance. In chronic myelogenous leukemia, treatment with imatinib and indomethacin reduced the self-renewal ability of CML stem cells by inhibiting prostaglandin-mediated stabilization of β-catenin [111].

Resveratrol, a phytophenol previously identified as a Wnt inhibitor, has been tested in patients with colorectal cancer and healthy ones. In a phase I clinical trial, low doses of resveratrol inhibited Wnt signaling in normal colonic mucosa but failed in cancer cells, suggesting a role in colorectal cancer prevention rather than treatment [112]. Other agents, such as salinomycin, an antibiotic potassium ionophore, have been shown to selectively kill breast cancer stem cells by blocking Wnt/β-catenin signaling and inducing degradation of the Wnt co-receptor LRP6 [113,114].

Although the focus on Wnt research has advanced significantly lately, no Wnt-targeted treatments have yet been approved. There are, however, several clinical trial underway for molecules like anti-Fzd antibodies, Fzd domain-containing recombinant proteins (NCT01608867), inhibitors of β-catenin/CBP interactions (NCT01606579), and Porcupine inhibitors (NCT01351103).

## 3. Conclusions

Many facts suggest that activated Wnt signaling plays a key role in the development of many human cancer types, driving tumor proliferation, aggressive behavior, and metastasis. Due to the complexity of Wnt signaling, studying and targeting this pathway remains challenging, and efforts to incorporate new Wnt-targeting strategies into clinical trials have been limited. The high cost of globally inhibiting Wnt signaling in the human body underscores the need for further research to develop selective and safe approaches for inhabiting specific components of the Wnt pathway or for targeting alternative pathways involved in crosstalk with Wnt signaling. Despite these challenges, novel anti-cancer strategies targeting aberrant Wnt signaling hold promise in the fight against cancer, one of humanity’s deadliest diseases.

## Figures and Tables

**Table 1 genes-15-01597-t001:** Wnt inhibitors under development.

Inhibitory Agent	Target/Mechanism	Phase/Status of Development
IWP Molecules	Inhibitors of Porcupine (Wnt ligand secretion)	Preclinical studies (effective in gastric, breast, and head/neck cancers in models)
Anti-Wnt-3a Antibodies	Target Wnt ligands	Preclinical (antitumoral effects in prostate cancer models)
Anti-Fzd-10 Antibodies	Target Frizzled-10 receptor	Preclinical (effective in synovial sarcoma; Yttrium-90 conjugation under exploration)
Recombinant Fzd Proteins	Soluble Wnt receptors (e.g., Fzd-8 extracellular domain)	Preclinical (effective in teratocarcinomas)
Dkk Proteins	Target LRP co-receptors (Wnt signaling inhibition)	Preclinical (induces apoptosis in multiple cancers, though some variants have pro-invasive effects)
Tankyrase Inhibitors (XAV939)	Stabilize Axin to promote β-catenin degradation	Preclinical (effective in APC-mutant colorectal cancer, but toxicity to normal cells is a concern)
Pyrvinium	Activates CK1 to enhance β-catenin degradation	Preclinical (potential against β-catenin-activated cancers)
Peptides Targeting Dvl-Fzd	Block Dvl-Fzd interactions	Preclinical (induce apoptosis selectively in non-small cell lung cancer, melanoma, and HCC cells)
Resveratrol	Wnt signaling inhibitor	Phase I (prevention role in colorectal cancer; no effect in cancer cells)
Salinomycin	Targets LRP6 and Wnt co-receptors	Preclinical (selectively kills breast cancer stem cells)
Anti-Fzd Antibodies	Target Frizzled receptors	Phase I/II (NCT01608867)
Porcupine Inhibitors	Block Wnt ligand secretion	Phase I (NCT01351103)
β-catenin/CBP Interaction Inhibitors	Block β-catenin transcriptional activity	Phase I (NCT01606579)

## Data Availability

No new data were created or analyzed in this study.

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
