# Peer review of "Wnt Signaling Pathway in Tumor Biology"

_genes, 2024, doi:10.3390/genes15121597_

Round 1
Reviewer 1 Report
Comments and Suggestions for Authors
Authors do an in-depth review about the importance of the wnt pathway in cáncer.
Main issues:
Hedgehog and notch are not mentioned in the crosstalk section although as stated in the introduction they have recognized interactions with Wnt. I would modify extensively the crosstalk section to include these important relationships. (line 37,“Various key signaling networks, mainly Hedgehog, Notch, and Wnt, appear to influence the fate and behavior of CSCs”)
In the treatment section, I would explain in more detail the phase and development of the wnt inhibitors currently under development.
Minor issues:
Line 60, This phrase does not make sense to me. Wnt/β-catenin, Notch, Hedgehog (Hh) are three major signaling pathways fundamentally involved in unraveling the complexities of cáncer
Line 129, phrases should not start with a number.” 15% of sporadic colon cancers…”. It happens again in line 204.
Line 136 “E-cadherin, a form of β-catenin located in adherent cell junctions” . Are the authors sure thay e-cadherin is a form of b-catenin? Please provide a reference.
Line 201 Desirably, CSCs-specific inhibitors should be developed to minimize the side effects on normal tissues. How do the authors this pathway could just be inhibited on CSC?
Line 226 “While the Wnt pathway is well-studies,”ammend please.
Line 274 “Among the proteins in the β-catenin degradation complex, APC, has the maximal capacity to inactivate mutations” This phrase does not make sense to me. Explain or please modify.
Line 291 “Restoration of miR-30 activity in stromal cells has been shown to reduce drug sensitivity in chemoresistant mielomas” reduce or increase?
Author Response
Review Genes
Comments and Suggestions for Authors
Authors do an in-depth review about the importance of the wnt pathway in cáncer.
Main issues:
Hedgehog and notch are not mentioned in the crosstalk section although as stated in the introduction they have recognized interactions with Wnt. I would modify extensively the crosstalk section to include these important relationships. (line 37,“Various key signaling networks, mainly Hedgehog, Notch, and Wnt, appear to influence the fate and behavior of CSCs”)
The following paragraph was added to the Crosstalk section: As Wnt signaling was described to have a pivotal role in tumorigenesis, it has be-come an attractive therapeutic strategy. However, there are numerous reports that Wnt/β-catenin cascade is often associated with other two signaling pathways: Hh, and Notch [53]. Cross-talk points between these three pathways involve mutual regulation whereby they can either activate or repress each other's activity in cancer. Notch specific ligand, Jagged1 was shown to downregulate Gli2, an Hh effector, and thus promote apoptosis and increase sensibility to drugs, as observed in a study conducted on in vitro and in vivo ovarian cancer models [54]. A separate study showed that Gli1, another Hh effector, was silenced by Hes1, a Notch-specific protein that encodes a basic helix–loop–helix (bHLH) transcriptional repressor, in glioblastoma neurosphere lines and primary human glioblastoma lines [55]. Wnt interacts with both Notch and Hh, as to stabilize Gli or repress β-catenin through Gli interactions. Acar et al, 2021, have brought strong ev-idence that Notch has a repressive effect on Wnt, by forming a complex with β-catenin, through two proteins, ΔEGF_N1 and NICD (Notch intracellular domain), thus blocking the transcription and expression of β-catenin [56]. This complements a previous study from 2014, where Collu et al correlated GSK3β, a member protein of the β-catenin de-struction complex, that additionally phosphorylates NICD and enables its translocation in the nucleus, for transcription activation [57]. Similarly, Wnt and Hh also work in an alternative manner. Hh enhances β-catenin stability, an activator of Wnt ligands ex-pression, while Wnt inactivates Hh through β-catenin degradation of gli proteins [58]. Collectively, these pathways promote processes like EMT, CSCs maintenance, and therapy resistance, illustrating a cooperative contribution to tumor progression and treatment evasion [6].
In the treatment section, I would explain in more detail the phase and development of the wnt inhibitors currently under development.
The following table was introduced in order to summaries the current wnt inhibitors under development:
|
Inhibitory agent |
Target/Mechanism |
Phase/Status of development |
|
IWP Molecules |
Inhibitors of Porcupine (Wnt ligand secretion) |
Preclinical studies (effective in gastric, breast, and head/neck cancers in models) |
|
Anti-Wnt-3a Antibodies |
Target Wnt ligands |
Preclinical (therapeutic effects in prostate cancer models) |
|
Anti-Fzd-10 Antibodies |
Target Frizzled-10 receptor |
Preclinical (effective in synovial sarcoma; Yttrium-90 conjugation under exploration) |
|
Recombinant Fzd Proteins |
Soluble Wnt receptors (e.g., Fzd-8 extracellular domain) |
Preclinical (effective in teratocarcinomas) |
|
Dkk Proteins |
Target LRP co-receptors (Wnt signaling inhibition) |
Preclinical (induces apoptosis in multiple cancers, though some variants have pro-invasive effects) |
|
Tankyrase Inhibitors (XAV939) |
Stabilize Axin to promote β-catenin degradation |
Preclinical (effective in APC-mutant colorectal cancer, but toxicity to normal cells is a concern) |
|
Pyrvinium |
Activates CK1 to enhance β-catenin degradation |
Preclinical (potential against β-catenin-activated cancers) |
|
Peptides Targeting Dvl-Fzd |
Block Dvl-Fzd interactions |
Preclinical (induce apoptosis selectively in non-small cell lung cancer, melanoma, and HCC cells) |
|
Resveratrol |
Wnt signaling inhibitor |
Phase I (prevention role in colorectal cancer; no effect in cancer cells) |
|
Salinomycin |
Targets LRP6 and Wnt co-receptors |
Preclinical (selectively kills breast cancer stem cells) |
|
Anti-Fzd Antibodies |
Target Frizzled receptors |
Phase I/II (NCT01608867) |
|
Porcupine Inhibitors |
Block Wnt ligand secretion |
Phase I (NCT01351103) |
|
β-catenin/CBP Interaction Inhibitors |
Block β-catenin transcriptional activity |
Phase I (NCT01606579) |
Minor issues:
Line 60, This phrase does not make sense to me. Wnt/β-catenin, Notch, Hedgehog (Hh) are three major signaling pathways fundamentally involved in unraveling the complexities of cancer
The sentence was rephrased: “Wnt/β-catenin, Notch, Hedgehog (Hh) are three major signaling pathways that play fundamental roles in understanding the complex mechanisms of cancer development, regulation, and progression.”
Line 129, phrases should not start with a number.” 15% of sporadic colon cancers…”. It happens again in line 204.
The phrases were corrected: “Around 15% of sporadic colon cancers have showed mutations in the β-catenin gene that make it resistant to degradation, leading to constitutive pathway activation.”
“A total of 19 Wnt ligands and 10 Frizzled receptors have been described so far, and the interactions between them are not uniform. For example, Wnt-5a and Wnt-7a can suppress oncogenic Wnt signaling by triggering non-canonical Wnt pathways.”
Line 136 “E-cadherin, a form of β-catenin located in adherent cell junctions” . Are the authors sure thay e-cadherin is a form of b-catenin? Please provide a reference.
E-cadherin is a different protein type from β-catenin, and the sentence was rephrased in a clearer manner: “EMT leads to the loss of E-cadherin, an important part of the of E-cadherin/β-catenin adhesion complex, a form of adherens cell-cell junction.”
Line 201 Desirably, CSCs-specific inhibitors should be developed to minimize the side effects on normal tissues. How do the authors this pathway could just be inhibited on CSC?
The following paragraph was added: Recently, novel approached have been proposed to achieve this goal. The development of multi-task nanoparticle was proposed, for a more specific target, with improved drug-dosing and also, possibility of drug combination cargo, for a more versatile application that would focus on at least one mechanism involved in cancers, unlike free drugs currently unsed in common therapies. Multiple models are under development with promising results, but still unclear regarding long-term toxicity and off-target events [70].
Line 226 “While the Wnt pathway is well-studies,”ammend please.
“well-studied” corrected.
Line 274 “Among the proteins in the β-catenin degradation complex, APC, has the maximal capacity to inactivate mutations” This phrase does not make sense to me. Explain or please modify.
The phrase has been modified: “Overexpressed β-catenin is phosphorylated and marked with ubiquitin for proteasomal digestion by a destruction complex. Among all the proteins included in this degradation assemble, namely APC, Axin, GSK-3β, and CK1, APC has a pivotal role in blocking excessive Wnt signaling, by facilitating the binding between β-catenin to the rest of the proteins in the complex.”
Line 291 “Restoration of miR-30 activity in stromal cells has been shown to reduce drug sensitivity in chemoresistant mielomas” reduce or increase?
Indeed, the drug sensitivity is increased. The text was corrected.
Reviewer 2 Report
Comments and Suggestions for Authors
Wnt signaling pathway in tumor biology
Comments:
The current review by Iluta S. and colleagues, titled “Wnt signaling pathway in tumor biology,” highlights the role of cancer stem cells (CSCs) in contributing to treatment resistance, with a particular emphasis on the Wnt pathway's involvement in the self-renewal and tumorigenic potential of CSCs. The authors thoroughly describe the molecular components of both canonical and non-canonical Wnt signaling pathways. They provide a detailed explanation of β-catenin degradation complex proteins, including APC, Axin, and GSK3β, whose mutations are implicated in various cancers. Additionally, the review discusses the role of Wnt signaling in metastasis, particularly its influence on epithelial-mesenchymal transitions. Finally, the authors explore the promising potential of targeting Wnt signaling for cancer treatment, while also addressing the associated challenges. To conclude, this review paper could be a great resource on Wnt siganling involvement in cancer. The following are few suggestions to improve the quality of the review:
1. The structure of the manuscript could be improved. For instance, while the title specifically focuses on Wnt, the abstract contains limited information about Wnt. Additionally, Section 1 (Wnt/β-catenin signaling and cancer development) begins discussing Wnt with subsections-1.1 and 1.2, however there is no standalone Section 2?
2. Since the review focuses on the Wnt pathway, the authors could have included a discussion on the origins of its discovery (Int-1), highlighting its initial identification as a mammalian oncogene and its subsequent role in regulating embryonic development.
3. As the review centers on Wnt and cancer, the authors could have included a specific paragraph discussing the role of the Wnt pathway in regulating the cell cycle.
Typos to fix:
Line 64: ‘Axis’ instead of ‘axix’
Line 79: When Wnt signaling ‘is’ instead of ‘in’
Line 226: Wnt pathway is ‘well-studied’ instead of ‘well-studies’
Author Response
Comments:
The current review by Iluta S. and colleagues, titled “Wnt signaling pathway in tumor biology,” highlights the role of cancer stem cells (CSCs) in contributing to treatment resistance, with a particular emphasis on the Wnt pathway's involvement in the self-renewal and tumorigenic potential of CSCs. The authors thoroughly describe the molecular components of both canonical and non-canonical Wnt signaling pathways. They provide a detailed explanation of β-catenin degradation complex proteins, including APC, Axin, and GSK3β, whose mutations are implicated in various cancers. Additionally, the review discusses the role of Wnt signaling in metastasis, particularly its influence on epithelial-mesenchymal transitions. Finally, the authors explore the promising potential of targeting Wnt signaling for cancer treatment, while also addressing the associated challenges. To conclude, this review paper could be a great resource on Wnt siganling involvement in cancer. The following are few suggestions to improve the quality of the review:
- The structure of the manuscript could be improved. For instance, while the title specifically focuses on Wnt, the abstract contains limited information about Wnt. Additionally, Section 1 (Wnt/β-catenin signaling and cancer development) begins discussing Wnt with subsections-1.1 and 1.2, however there is no standalone Section 2?
The abstract was modified and so was the structure of the article, according to the reviewer’s suggestions.
- Since the review focuses on the Wnt pathway, the authors could have included a discussion on the origins of its discovery (Int-1), highlighting its initial identification as a mammalian oncogene and its subsequent role in regulating embryonic development.
The following paragraphs were added:
Wnt pathway was first discovered in 1982, by Nusse et al, as an attempt to inves-tigate the genetic modifications involved in the development of breast cancer in mice. The infection of C3H murine models with the mouse mammary tumor virus (MMTV) led to the observation of a proviral insertion mutation in chromosome 15, at a specific locus, which led to the first description of the int-1 (integration 1) gene, an oncogene activated by the retroviral infection. At the same time, independent studies in Drosophila melanogaster elucidated the wingless (wg) gene as the core component of embryonic segment polarity. Further investigations linked wg with int-1 as their homologous genes and came up with the collective term Wnt-"wingless" undoing the contrived criteria that initally coined "in1" into int for naming.
Then it became active in embryogenesis when its role was played in regulating body anterior-posterior axis formation, enabling of epithelial to mesenchymal transition of cells during gastrulation, organogenesis, cell fate specification, and tissue patterning in development. For example, in lung physiological development, the inhibition of the Wnt/β-catenin pathway impacts the differentiation of proximal stem cells into airway epithelial cells, while it’s activation results in distal epithelial cells specialization. In the skeletal system, Wnt signaling affects the proliferation and differentiation of mes-enchymal stem cells and osteoclasts, contributing to bone formation, and maintenance of mass and strength. Wnt pathway also influences the development of the neural tube, heart and kidney, through mechanisms that sustain and enable the proper population of stem cells to differentiate into the mature and functional cells and tissues.
- As the review centers on Wnt and cancer, the authors could have included a specific paragraph discussing the role of the Wnt pathway in regulating the cell cycle.
The following paragraph was introduce to discuss the involvement of Wnt pathway in cell cycle:
With the later discoveries that underpinned Wnt, a stimulating link with the cell cycle had been established because, through β-catenin stabilization, it influences the transcription and expression of key regulatory genes like cyclin D1 and c-myc, essential for the G1/S transition in the cell cycle [14] In the case of pluripotent stem cells, Wnt signailng involved in different steps of the cell cycle. First, just like in the case of somatic cells, it promotes cell proliferation by facilitating G1/S transition, due to cyclin D1 transcription and inhibition of p21 and p27. Proliferation is also stimulated by activation of the highly conserved cyclin Y, with indispensable role in mitosis regulation (G2/M transition).
Typos to fix:
Line 64: ‘Axis’ instead of ‘axix’
The typo was corrected.
Line 79: When Wnt signaling ‘is’ instead of ‘in’
The phrase was modified.
Line 226: Wnt pathway is ‘well-studied’ instead of ‘well-studies’
The typo was corrected.
Round 2
Reviewer 1 Report
Comments and Suggestions for Authors
Minor and major changes have been done.